# Overexpression of *GmRIQ2-like* (*Glyma.04G174400*) Enhances the Tolerance of Strong Light Stress and Reduces Photoinhibition in Soybean (*Glycine max* (L.) Merr.)

**Jing Deng [†], Dongmei Li [†], Huayi Yin, Li Ma, Jiukun Zhang and Binbin Zhang ***

Key Laboratory of Soybean Biology in Chinese Education Ministry (Northeastern Key Laboratory of Soybean Biology and Genetics &Breeding in Chinese Ministry of Agriculture), Northeast Agricultural University, Harbin 150030, China; 13199075975@163.com (J.D.); yy841056@163.com (D.L.); yhy9762@163.com (H.Y.); mail2020mailbox@163.com (L.M.); zhang170301066@163.com (J.Z.)

**\*** Correspondence: zhangbinbin@neau.edu.cn; Tel.: +86-186-4609-2893

**†** These authors contributed equally to this work.

**Abstract:** Soybean (*Glycine max* (L.) Merr.) is an important crop that serves as a source of edible oil and protein. However, little is known about its molecular mechanism of adaptation to extreme environmental conditions. Based on the *Arabidopsis thaliana* sequence database and Phytozome, a soybean gene that had a highly similar sequence to the reduced induction of the non-photochemical quenching2 (*AtRIQ2*) gene, *GmRIQ2-like* (accession NO.: *Glyma.04G174400*), was identified in this study. The gene structure analysis revealed that *GmRIQ2-like* encoded a transmembrane protein. Elements of the promoter analysis indicated that *GmRIQ2-like* participated in the photosynthesis and abiotic stress pathways. The subcellular localization results revealed that the protein encoded by *GmRIQ2-like* was located in chloroplasts. The quantitative real-time (qRT)-PCR results revealed that *GmRIQ2-like*-overexpression (OE) and -knock-out (KO) transgenic soybean seedlings were cultivated successfully. The relative chlorophyll (Chl) and zeaxanthin contents and Chl fluorescence kinetic parameters demonstrated that *GmRIQ2-like* dissipated excess light energy by enhancing the non-photochemical quenching (NPQ) and reduced plant photoinhibition. These results suggested that *GmRIQ2-like* was induced in response to strong light and depressed Chl production involved in soybean stress tolerance. These findings indicate that the transgenic seedlings of *GmRIQ2-like* could be used to enhance strong light stress tolerance and protect soybean plants from photoinhibition damage. This study will serve as a reference for studying crop photoprotection regulation mechanisms and benefits the research and development of new cultivars.

**Keywords:** soybean (*Glycine max* (L.) Merr.); NPQ; photoinhibition; bioinformatics; subcellular localization; qRT-PCR

---

## 1. Introduction

Plants absorb light energy through chloroplasts located in their photosynthetic organs [1], which is subsequently converted into stable chemical energy and stored in inorganic substances that are synthesized into organic substances, and oxygen is released [2,3]. Light energy is an indispensable factor for plant growth and development [4–7]. When light energy is insufficient, plants regulate their external structures to capture enough light energy necessary for growth and development [8,9]. When light energy is in excess, it is absorbed by plants and exceeds their utilization ability, which results in photoinhibition [10–12]. When photosynthesis is inhibited, photosynthetic efficiency and capacity

are reduced, and excess light energy that cannot be dissipated accumulates in the photosystem, thereby forming active oxygen [13–15], and causing photooxidation and photodamage in serious cases, which eventually leads to plant death [16–18].

Photoinhibition is a common phenomenon in nature [3]. When strong light simultaneously occurs with drought, heat, cold, salinity, nutrient deficiency or other stressors, photoinhibition is exacerbated [19–24]. A previous study demonstrated that drought stress induced the regulatory energy dissipation quantum yield of *Kobresia alpinensis* and *Populus euphratica* as light intensity increased, resulting in increased photoinhibition [25]. Soybean (*Glycine max* (L.) Merr.) is a main oil grain crop that often suffers from a variety of abiotic stressors [26]; thus, the degree of photoinhibition in this crop is relatively serious [27]. However, over their evolutionary histories, plants have formed a variety of defense mechanisms to protect their photosynthetic organs [28–30], including chloroplasts that avoid light movement, heat dissipation, active oxygen removal, and transition of the photosystem [6,31–33].

Nevertheless, superfluous photons that exceed the plant's capacity could be deleterious [34]. Moreover, redundant light energy leads to the inhibition of photosynthetic carbon fixation [35–37]. Under light stress, as the chlorophyll (Chl) content and oxygen evolution capacity decrease, the thylakoid structure is damaged [27,38]. Excess excitation energy produced by strong light reduces photosynthetic electron transfer, which combines with oxygen to form active oxygen [9,39]. Active oxygen destroys photosynthetic organs and reaction centers (RCs), resulting in reduced net photosynthetic rates and photochemical efficiency of Photosystem II (PS II) [40]. The reduction of the photosynthetic efficiency of PS II results in photoinhibition [41]. Under high light intensity, excess light energy that was absorbed is dissipated as heat [42].

RIQ proteins link the grana structure and organize light-harvesting complex II proteins (LHC IIs) [42], which play important roles in grana biogenesis [43–46]. PS II is located in thylakoid membranes of the chloroplasts in land plants, which mainly contain RCs, Chl *a/b* LHC IIs, and an oxygen-evolving complex (OEC) [47,48]. Excessive Chl contents in the leaves make plants more sensitive to photoinhibition [49]. High light intensity stimulates the protection mechanism of plants, leading to higher non-photochemical quenching (NPQ) coefficients [50]. Higher NPQ induction levels and lower Chl contents balance the absorption of light energy under strong light conditions, thereby enhancing the adaptability of leaves in strong light environments [51]. Therefore, regulating the expression of proteins encoded by RIQ may provide a new method for improving the ability of plants to withstand environmental stressors.

In this study, a strong light-induced *GmRIQ2-like* gene was isolated from soybeans to stabilize the genetic overexpression (OE) and knockout (KO) of transgenic soybean seedlings as materials, and we explored the role of NPQ in light energy absorption to identify potential regulatory networks under light stress conditions. This gene effectively led to transgenic soybean plants that avoided photoinhibition and photodamage, providing experimental evidence for the improvement of soybean resistance to strong light stress. These findings will serve as a useful reference for the breeding of soybeans resistant to stress.

## 2. Materials and Methods

### 2.1. Bioinformatics Analysis

The *AtRIQ2* protein sequence was obtained from TAIR [52]. Soybean DNA, mRNA, coding regions (CDS), and protein sequences of *GmRIQ2-like* were BLASTed using *AtRIQ2* in Phytozome [53]. The physicochemical properties of *GmRIQ2-like* were observed using ExPasy [54]. TMHMM [55] was used to analyze the transmembrane. Promoter elements were analyzed using PlantCARE [56]. Phylogenetic analyses were conducted using MEGA5.0 (The Biodesign Institute, Tempe, AZ, USA). The correlated sequences of various species were acquired using BLASTp in the NCBI database [57] with the protein sequence encoded by *GmRIQ2-like*. The CRISPR/cas9 target sites of *GmRIQ2-like* were predicted using CRISPRdirect [58].

## 2.2. Plant Materials, Growth Conditions, and RNA Isolation

Soybean seeds of the "KenFeng 16" and "DongNong 50" cultivars were provided by the Soybean Research Institute of Northeast Agricultural University (Harbin, China). "KF16" seedlings were grown in soil:vermiculite (1:1) for 3 weeks under a 16/8 h light/dark photoperiod at 23 °C. The second stage of fully expanded trifoliate leaves was harvested and stored at 80 °C for RNA isolation. Total RNA was extracted from "KF16" seedlings using a Trizol kit (TAKARA, Shiga, Japan) following the manufacturer's instructions. First-strand cDNA was synthesized using first-strand cDNA synthesis (Tiangen, Beijing, China) for PCR amplification.

## 2.3. Construction of Related Vectors

Full-length CDS of *GmRIQ2-like* was amplified using PrimeSTAR GXL DNA polymerase (TAKARA, Shiga, Japan). The PCR amplification procedure was as follows: 94 °C for 5 min; 94 °C for 30 s, 55 °C for 30 s, and 35 cycles at 72 °C for 30 s; and 72 °C for 10 min. The PCR products were cloned into the pGM-T vector (Tiangen, Beijing, China). Afterward, the ligation products were transformed into *E. coli* competence trans-T1 (Tiangen, Beijing, China). Single colonies were sequenced by Sangon Biotech (Beijing, China). Single enzyme cutting of the OE vector, pCXSN-bar, was conducted following the methods described by Li et al. [59] and performed by *XcmI*. Ligation between the CDS of *GmRIQ2-like* and pCXSN-bar was performed using T$_4$ ligase (Tiangen, Beijing, China). Finally, the *GmRIQ2-like*-OE recombinant vector (*GmRIQ2*-a) was transformed into *Agrobacterium tumefactions* EHA105 using the freeze-thawing method [60]. The primers used for PCR of *GmRIQ2-like* cloning are provided (Table 1).

**Table 1.** List of primers used in this study.

| Primers | Primer Sequences (5′–3′) |
| --- | --- |
| *GmRIQ2-like*-OE-S | 5′-GCAACCTCGTCCCAACAAC-3′ |
| *GmRIQ2-like*-OE-A | 5′-CACCACCGAACCACCCATT-3′ |
| *GmRIQ2-like*-qRT-S | 5′-CCATTGTGCTCATTCCCG-3′ |
| *GmRIQ2-like*-qRT-A | 5′-ACCACTTCGCAACCATCT-3′ |
| *Actin4*-S | 5′-GTGTCAGCCATACTGTCCCCATT-3′ |
| *Actin4*-A | 5′-GTTTCAAGCTCTTGCTCGTAATCA-3′ |
| *GmRIQ2-like*-GFP-S | 5′-ATGGCGGTCCCTGCTACATCCTCA-3′ |
| *GmRIQ2-like*-GFP-A | 5′-GCAAGCCGCCGAGCACGAC-3′ |
| T7-gRNA-FPg | 5′-TAATACGACTCACTATAGGAGAAGGAGTGATTGGTGGTTTTAGTACTCTGGAAACAG-3′ |
| gRNA-RP | 5′-ATCTCGCCAACAAGTTGACGAG-3′ |
| CRISPR-*GmRIQ2-like*-S | 5′-TTGGAGAAGGAGTGATTGGTG-3′ |
| CRISPR-*GmRIQ2-like*-A | 5′-AACCACCAATCACTCCTTCTC-3′ |
| CRISPR-seq-A | 5′-CTCCTTCCTTCCGTCCACTTCATC-3′ |

Bgl II and Spe I double enzyme digestion (TAKARA, Shiga, Japan) of pCAMBIA1302 was employed for the *Arabidopsis* protoplast fluorescent localization expression vector. The cloning and connection processes are described above. The primers used for PCR of fluorescence localization are provided (Table 1).

The gene KO target site efficiency detection and CRISPR/cas9 vector construction of *GmRIQ2-like* were performed using a saCas9-gRNA target site efficiency detection kit (Viewsolid Biotech, Beijing, China) and a Plant Cas9/gRNA plasmid construction kit (Viewsolid Biotech, Beijing, China) following the manufacturer's instructions. By predicting the target sites of *GmRIQ2-like* exons, three sites were identified: GAGAAGGAGTGATTGGTGCGG, CCATGGCGAAGACGGTTAGGG, and AGAAGGAGTGGTTTGGAG. The activity of the GAGAAGGAGTGATTGGTGCGG target sites comprised up to 85% of the total according to the enzyme digestion assay and was selected for CRISPR/cas9 vector construction. The primers used for the CRISPR/cas9 technique are provided (Table 1).

## 2.4. Subcellular Localization in Arabidopsis Protoplasts

*GmRIQ2-like*-GFP plasmids were used for transient assays via the polyethylene glycol transfection of *Arabidopsis* protoplasts, as described by Yoo et al. [61]. Transfected cells were imaged using a TCS SP2 confocal spectral microscope imaging system (Leica, Wetzlar, Germany).

## 2.5. Cultivation of Transgenic Soybeans

The transformation of transgenic *A. tumefaciens* into 'DN50' cotyledon nodes was conducted following the methods described by Jia et al. [62] with minor modifications. The reagents used for transformation are provided (Table 2). 'DN50' seeds were sterilized using chlorine and placed on germination medium (GM). *A. tumefaciens* (50 μL) expressing the vectors *GmRIQ2*-a or *GmRIQ2*-b was added to 50 mL YEP liquid. The solutions were centrifuged for 10 min at 4000 rpm until the OD value was 0.6–0.8. Then, the scratched cotyledons were placed on transfection medium (LCCM) for 3 d. Afterward, the cotyledons were recovered on shoot-induced medium (SCCM⁻) for 7 d. Then, the cotyledons were filtered on selected medium (SCCM⁺) with 5 mg/L phosphinothricin (PPT) for 7 d. Resistant cluster buds were placed in elongated medium (SEM) for 15–20 d until the buds were 3–4 cm long. Buds were cut and transferred to root media (RM) when the main roots were 3–5 cm long. Finally, seedlings were transplanted into soil:vermiculite (3:1) under a 16/8 h light/dark photoperiod at 23 °C until the seedlings were mature.

**Table 2.** Media used for soybean cotyledonary explant transformation.

| Solution | pH | Composition (g/L) |
|:---:|:---:|:---:|
| GM | 5.8 | B5: 3.21 g; Sucrose: 20 g; Agar: 7 g; 6-BA: 1 mg/L |
| LCCM | 5.4 | B5: 0.321 g; Sucrose: 30 g; Mes: 3.9 g; Acetosyringone (As): 160 mg/L;Gibberellic acid (GA): 10 mg/L; 6-BA: 1 mg/L |
| SCCM | 5.4 | B5: 0.321 g; Sucrose: 30 g; Mes: 3.9 g; Agar: 5 g; As: 160 mg/L; GA: 10 mg/L;BA: 1 mg/L; DTT: 0.154 g; L-cysteine: 1 g; $Na_2S_2O_3$: 0.248 g |
| SSIM⁻ | 5.6 | B5: 3.21 g; Sucrose: 30 g; Mes: 0.59 g; Agar: 7.5 g; 6-BA: 1 mg/L; Cefotaxime (cef): 250 mg/L |
| SSIM⁺ | 5.6 | B5: 3.21 g; Sucrose: 30 g; Mes: 0.59 g; Agar: 7.5 g; 6-BA: 1 mg/L;Phosphinothricin (PPT): 5 mg/L; cef: 250 mg/L |
| SEM | 5.6 | Murashige and Skoog (MS): 4.43 g; Sucrose: 30 g; Mes: 0.59 g; Agar: 7.5 g;Aspirin (Asp): 50 mg/L; N,6-isopentenyladenine (ZR): 1 mg/L; Gibberellin ($GA_3$): 1 mg/LIndole-3-acetic acid (IAA): 1 mg/L; cef: 250 mg/L |
| RM | 5.6 | B5: 3.21 g; Sucrose: 20 g; Mes: 0.59 g; Agar: 7.0 g; Indole-3-butytric acid (IBA): 1 mg/mL |

Annotate: Germination medium (GM); Transfection medium (LCCM); Shoot-induced medium (SCCM⁻); Selected medium (SCCM⁺); Elongated medium (SEM); Root media (RM).

## 2.6. Detection of Transgenic Soybeans Using Quantitative Real-Time PCR

Total RNA was isolated from transgenic and wild-type (WT) soybean leaves using TRIzol reagent (TAKARA, Shiga, Japan). First-strand cDNA synthesis was performed as described above. Quantitative real-time (qRT)-PCR was performed on a Chromo4 RT-PCR system (Bio-Rad, Hercules, CA, USA) using SYBR Green PCR Master mix (Tiangen, Beijing, China). The soybean *actin4* gene was used as an internal standard control. The primers used for qRT-PCR are provided (Table 3). The PCR procedure was as follows: 95 °C for 5 min; 95 °C for 30 s, 60 °C for 30 s, and 40 cycles at 72 °C for 30 s; and 72 °C for 10 min. After qRT-PCR was conducted, the relative mRNA ratios were calculated using the $2^{-\Delta\Delta CT}$ method. All PCR experiments were conducted three times using prepared $T_2$ and $T_3$ generation homozygous independent cDNA. *GmRIQ2-like*-OE and -KO transgenic soybean seedlings identified as positive were named *GmRIQ2*-a1,a2,a3 and *GmRIQ2*-b1,b2,b3.

**Table 3.** Promoter elements of *GmRIQ2-like*.

| Name | Number | Function |
|---|---|---|
| G-Box | 8 | |
| box-4 | 5 | |
| box II | 2 | |
| ATCT-motif | 1 | |
| CATT-motif | 1 | involved in light responsiveness |
| CG-motif | 1 | |
| ACE | 1 | |
| GATA-motif | 1 | |
| I-box | 1 | |
| ABRE | 5 | involved in abscisic acid responsiveness |
| HSE | 3 | involved in heat stress responsiveness |
| TCA-element | 2 | involved in salicylic acid responsiveness |
| AuxRR-core | 1 | involved in auxin responsiveness |
| TC-rich repeats | 1 | involved in defense and stress responsiveness |
| circadian | 1 | involved in circadian control |
| ARE | 1 | essential for anaerobic induction |

### 2.7. Measurements of Physiological Characteristics in Transgenic Agronomic Traits

Soil-plant analyses development (SPAD) values were employed for detecting the relative Chl contents of transgenic soybean seedling leaves using a SPAD-502 Chl meter (Konica Minolta, Tokyo, Japan). First, the light intensity reading of $A_{650}$ and $A_{940}$ was recorded under non-blade conditions. After the blades were inserted, light intensity was read at the two wavelengths again. Finally, SPAD values were calculated using the following formula: SPAD = Klg (IRt/IR0)/(Rt/R0), where K is a constant, IRt is the infrared intensity of leaves under 940 nm, IR0 is the infrared intensity of non-blade parts under 940 nm, Rt is the infrared intensity of leaves under 650 nm, and R0 is the infrared intensity of non-blade parts under 650 nm.

The second stage of fully expanded trifoliate leaves was harvested and stored at −80 °C for zeaxanthin extraction. Then, the leaves were ground into a homogenate with 80% acetone and centrifuged at 200 g for 5 min. Finally, using 80% acetone as a reference, the light absorption of the liquid supernatant was detected at 505 and 652 nm (the light absorptions of Chl). Because the diurnal variation of the Chl contents was not obvious, the $A_{505}/A_{652}$ values were used to express the relative zeaxanthin contents of transgenic and WT soybean leaves.

Agronomic traits, including the plant height and number of main nodes, pods, and seeds of transgenic and WT plants, were measured based on grain physiological maturity dates. Transgenic and WT plants were grown in soil:vermiculite (1:1) pots under a 16/8 h light/dark photoperiod at 23 °C. The light intensity was maintained at about 500 μmol m$^{-2}$·s$^{-1}$ during the measurement.

### 2.8. Determination of Chl Fluorescence Kinetic Parameters

The Chl fluorescence kinetic parameters of the third fully expanded trifoliate leaves of the $T_2$ and $T_3$ generation transgenic 'DN50' control plants were measured using a Li-6400XT portable photosynthesis analyzer (Li-Cor, Lincoln, NE, USA). Leaves were dark-adapted for 30 min, and the initial fluorescence (Fo) under dark adaptation was measured with weakly-modulated measuring light. Then, a saturated flash (8000 μmol m$^{-2}$·s$^{-1}$, 0.8 s pulse time) was used to determine the maximum fluorescence (Fm) followed by continuous light activation (1000 μmol m$^{-2}$·s$^{-1}$) to irradiate the leaves in order to reach steady-state fluorescence (Fs). Afterward, a saturated flash (8000 μmol m$^{-2}$s$^{-1}$, 0.8 s pulse time) was used to determine the maximum fluorescence (Fm') under light adaptation. Finally, the photochemical light was turned off, and far-red light was turned on for 3 s to measure the minimum fluorescence (Fo') under light adaptation. The Fo, Fm, and PS II maximum photochemical efficiency (Fv/Fm), and NPQ, were obtained by recording measurements and calculations under 500 and 1800 μmol m$^{-2}$·s$^{-1}$

light intensity. Each treatment was repeated three times. For determination of the fluorescence kinetic parameters, refer to Li and Tao [63,64]. The parameters were calculated as follows: NPQ = Fm/Fm′-1. After testing, one of the third trifoliate leaves of the corresponding test leaf was retrieved, and the Chl fluorescence quenching process was measured indoors. After leaves were dark-adapted for 30 min, they were placed in a Chl fluorescence imager (IMAG-MAX/L, Walz, Germany), the application was implemented, the fluorescence parameter values were measured, and the images were saved.

*2.9. Statistical Analyses*

All results were repeated three times. Data are presented as the mean ± standard error (SE). The data were analyzed by Student's *t* test using SPSS v17.0 (SPSS Inc., Chicago, IL, USA).

## 3. Results

*3.1. Gene Structure and Phylogenetic Analyses of GmRIQ2-like*

According to the *AtRIQ2* protein sequence (TAIR: *AT1G74730*) in *Arabidopsis thaliana*, the full-length genomic sequence of *Glyma.04G174400*, named *GmRIQ2-like* in this study, was obtained using the BLAST tool in Phytozome (Figure 1A). The genomic, transcription, and CDS regions of *GmRIQ2-like* contained 2482, 1243, and 585 bp, respectively. Two introns were located in the CDS and 3′ untranslated regions with lengths of 688 and 18 bp, respectively. Proteins encoded by *GmRIQ2-like* consisted of 194 amino acids with a theoretical isoelectric point (pI) of 9.10 and estimated molecular weight of 20,385.7 Da. Additionally, the transmembrane structure was identified in the protein encoded by *GmRIQ2-like*, indicating that *GmRIQ2-like* was a transmembrane protein. The promoter elements analysis revealed that *GmRIQ2-like* participated in photosynthesis and various abiotic stress responses (Table 3). Specifically, there were 21 elements related to light responsiveness, including 5 ABRE, 3 HSE, 2 TCA, 1 AuxRR-core, TC-rich repeats, and ARE elements, which are involved in abscisic acid, heat, salicylic acid, auxin, defense, and stress responses, as well as anaerobic induction. To evaluate the evolutionary relationships within the *GmRIQ2-like* family, a combined phylogenetic analysis was performed using 14 RIQ2 proteins (Figure 1B). The results indicated that *GmRIQ2-like* was highly homologous with *Phaseolus vulgaris* L.

*3.2. Subcellular Localization of GmRIQ2-like*

*GmRIQ2-like* was predicted to serve a function related to NPQ. A potential protein encoded by *GmRIQ2-like* was found to be located in the chloroplasts. Therefore, the *GmRIQ2-like::GFP* fusion protein vector was transferred into *A. thaliana* protoplasts with high concentrations of PEG4000. The control protoplast was not transferred to the vector. Observations using a confocal laser scanning microscope revealed that the green fluorescence of *GmRIQ2-like::GFP* was located in the chloroplasts, while the control protoplast exhibited no fluorescence (Figure 2).

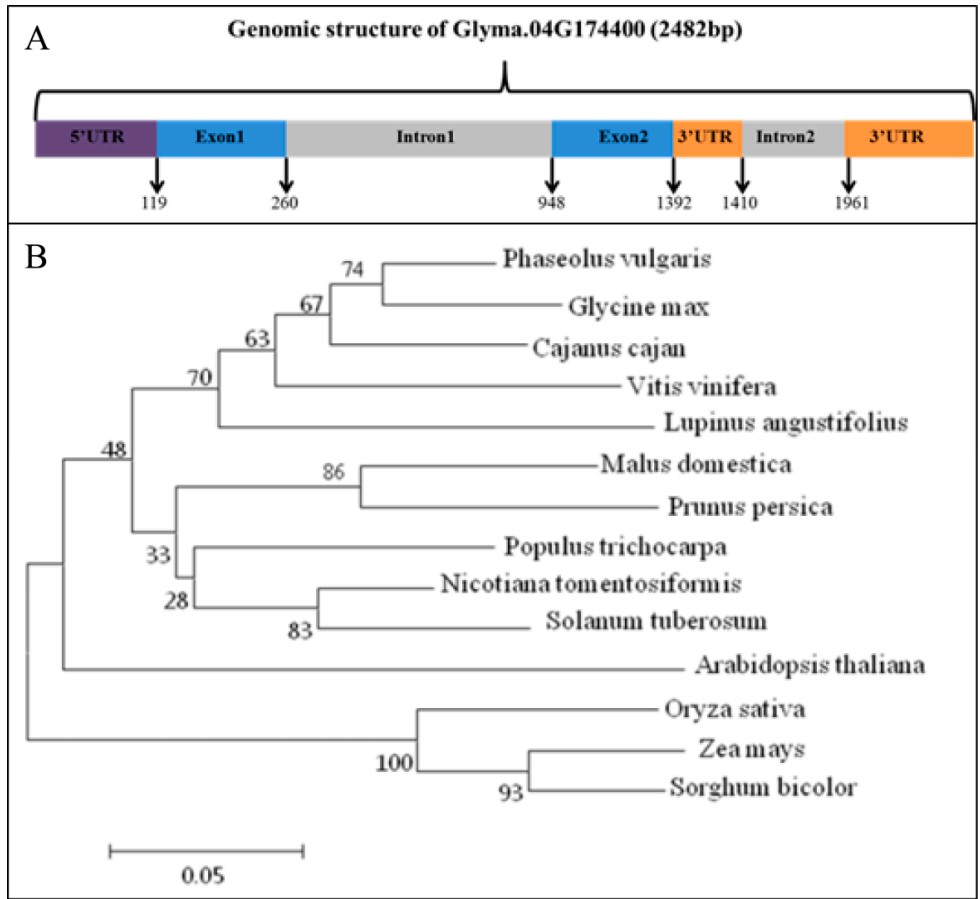

**Figure 1.** Bioinformatic analyses of *GmRIQ2-like*. (**A**) Genomic sequence structure of *GmRIQ2-like*. (**B**) Phylogenic analysis of *GmRIQ2-like*, which involved 14 amino acid sequences. In the phylogenetic tree, taxon names were replaced by the species name. Evolutionary analyses were conducted in MEGA5.0.

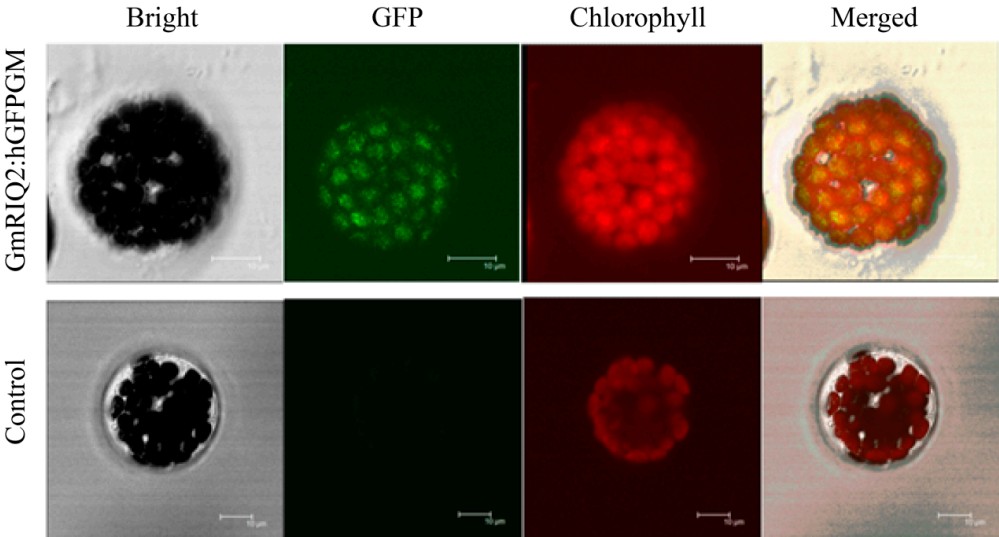

**Figure 2.** The subcellular localization of *GmRIQ2-like* was investigated in *Arabidopsis* protoplasts under a confocal microscope. The fluorescent distribution of control and fusion proteins of *GmRIQ2-like*–GFP were observed under white, UV, and red light. Bars = 10 μm.

### 3.3. Identification of GmRIQ2-like Transgenic Soybeans

The *GmRIQ2-like* overexpression vector (*GmRIQ2-a*) and *GmRIQ2-like* knock-out vector (*GmRIQ2-b*) were constructed successfully from the cDNA of soybean cultivar 'KF16' and transferred into 'DN50' soybean cotyledons by the Agrobacterium tumefaciens-mediated method. After the $T_1$ generation seeds of soybean were harvested, the quantitative real-time PCR (qRT-PCR) was used to detect positive transgenic seedlings (Figure 3). *GmRIQ2-like*-OE-transgenic soybean seedlings (a3) exhibited significantly higher expression levels, which were 14× higher than WT and greater than the other 6 detected seedlings (a1–a7) (Figure 3A). Additionally, seedlings a1 and a7 also exhibited various degrees of OE levels, which were 3.5 and 4× higher than WT (Figure 3A). The expression levels of *GmRIQ2-like* in all 6 detected KO transgenic soybean seedlings (b1–b6) were ≤1/5× lower than WT (Figure 3B). These results indicated that *GmRIQ2-like*-OE and -KO transgenic soybean seedlings were obtained successfully.

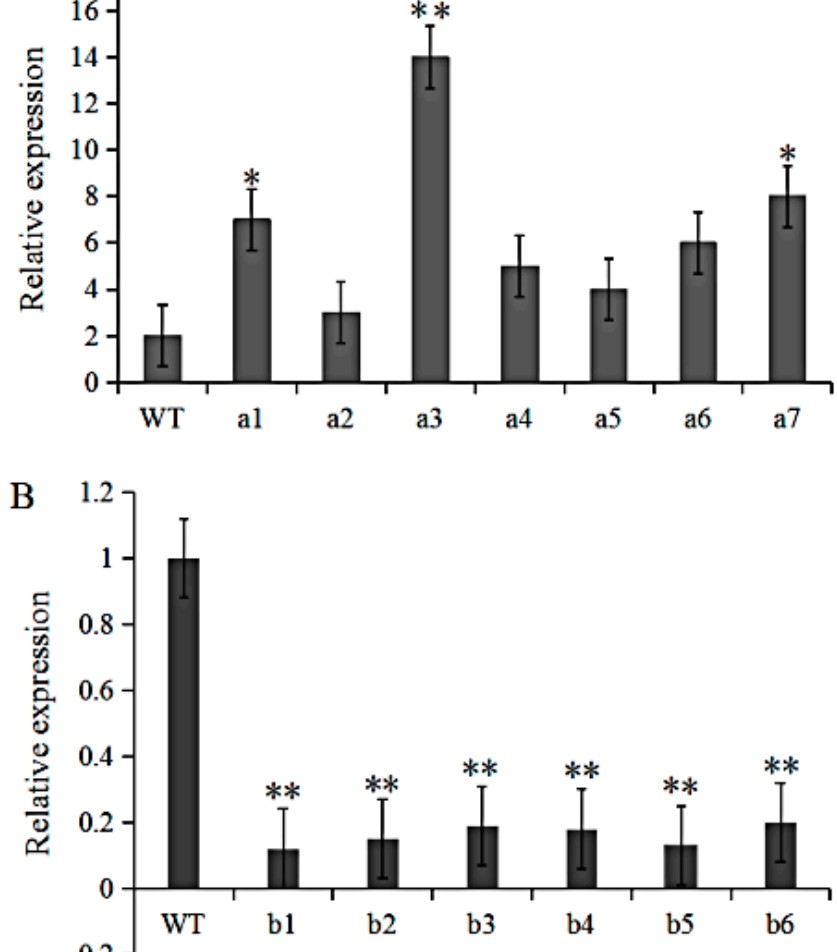

**Figure 3.** Identification of *GmRIQ2-like*-overexpression (OE) and -knock-out (KO) transgenic soybean seedlings. (**A**) Expression patterns of the $T_0$ generation of *GmRIQ2-like*-OE transgenic soybean seedlings (a1–a7). (**B**) Expression patterns of the $T_0$ generation of *GmRIQ2-like*-KO transgenic soybean seedlings (b1–b6). Relative transcript levels were determined by qRT-PCR using the $2^{-\Delta\Delta Ct}$ method; the *actin4* gene was used as an internal control. Data are presented as the average of three independent biological experiments ± SE. Asterisks indicate significant differences (* $p < 0.05$; ** $p < 0.01$) compared to the control.

### 3.4. Agronomic Traits and Physiological Indices of GmRIQ2-like Transgenic Seedlings

The agronomic traits of the $T_2$ generation of *GmRIQ2-like*-OE, -KO, and WT seedlings were measured (Table 4A). The plant height and number of nodes, pods, and seeds of the detected seedlings were ranked as follows: *GmRIQ2-like*-KO > WT > *GmRIQ2-like*-OE. Combined with the growth phenotypes (Figure 4A), these results indicated that the expression of *GmRIQ2-like* had a negative effect on soybean yield under the tested conditions.

**Table 4.** (**A**) Agronomic traits of the wild-type (WT) and $T_2$ generation of *GmRIQ2-like*-OE and -KO transgenic seedlings. (**B**) Agronomic traits of the WT and $T_3$ generation of *GmRIQ2-like*-OE and -KO transgenic seedlings.

| (A) | | | |
|---|---|---|---|
| **Agronomic Traits** | **WT** | ***GmRIQ2-like*-OE** | ***GmRIQ2-like*-KO** |
| Plant height (cm) | 68.5 ± 3.2 | 64.8 ± 2.4 | 71.0 ± 4.8 |
| Number of main nodes (no.) | 15 ± 2 | 15 ± 1 | 16 ± 2 |
| Number of pods (no.) | 27 ± 3 | 26 ± 2 | 28 ± 3 |
| Number of seeds (no.) | 71 ± 4 | 65 ± 3 | 74 ± 5 |
| Weight of seed per plant (g) | 6.3 ± 0.5 | 5.3 ± 0.4 | 6.9 ± 0.7 |
| Hundred-grain weight (g) | 8.9 ± 0.2 | 8.1 ± 0.2 | 9.3 ± 0.3 |
| (B) | | | |
| **Agronomic Traits** | **WT** | ***GmRIQ2-like*-OE** | ***GmRIQ2-like*-KO** |
| Plant height (cm) | 67.8 ± 2.8 | 62.7 ± 2.5 | 69.3 ± 3.2 |
| Number of main nodes (no.) | 15 ± 2 | 14 ± 2 | 16 ± 1 |
| Number of pods (no.) | 26 ± 1 | 25 ± 2 | 29 ± 2 |
| Number of seeds (no.) | 68 ± 3 | 66 ± 2 | 73 ± 4 |
| Weight of seed per plant (g) | 5.9 ± 0.5 | 5.5 ± 0.3 | 6.7 ± 0.7 |
| Hundred-grain weight (g) | 8.7 ± 0.3 | 8.4 ± 0.2 | 9.2 ± 0.4 |

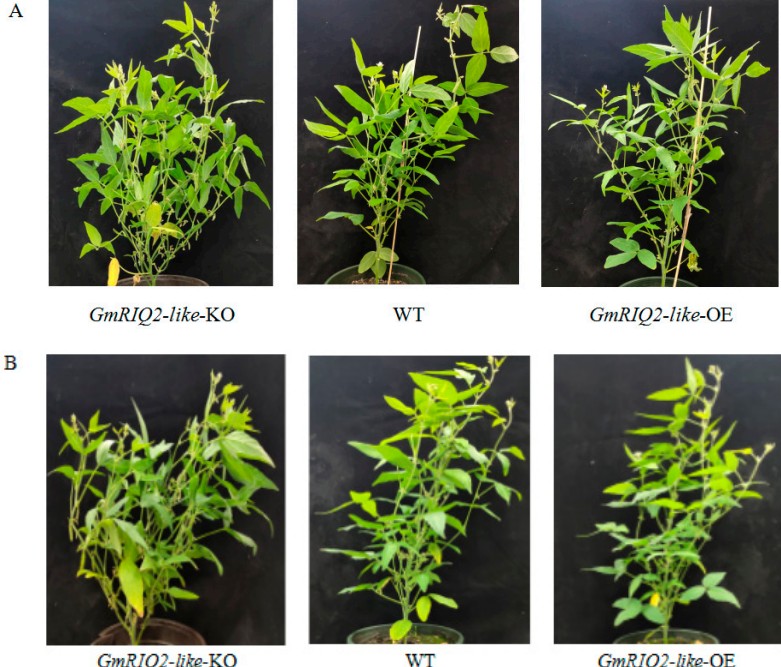

Annotate: The plant with Dongnong 50 as a control was named wild-type (WT).

**Figure 4.** Phenotypes of two-month old *GmRIQ2-like*-KO, -OE, and WT transgenic soybean seedlings. (**A**) Phenotype of the $T_2$ generation of transgenic soybean seedlings. (**B**) Phenotype of the $T_3$ generation of transgenic soybean seedlings. For each strain, 30 seedlings were selected.

Interestingly, under the tested light conditions, the SPAD values of the $T_2$ generation of *GmRIQ2-like*-OE seedlings were lower than WT (Figure 5A). The $A_{505}/A_{652}$ values in the $T_2$ generation leaves of *GmRIQ2-like*-OE, -KO, and WT seedlings were ranked as follows: *GmRIQ2-like*-OE > WT > *GmRIQ2-like*-KO (Figure 5B). The SPAD values, $A_{505}/A_{652}$ values, and agronomic traits of the $T_3$ generation of *GmRIQ2-like*-OE, -KO, and WT seedlings exhibited the same trends as the $T_2$ generation (Table 4B; Figure 6). These results further suggested that *GmRIQ2-like* enhanced NPQ, reduced Chl synthesis, and ultimately had some impact on soybean yield.

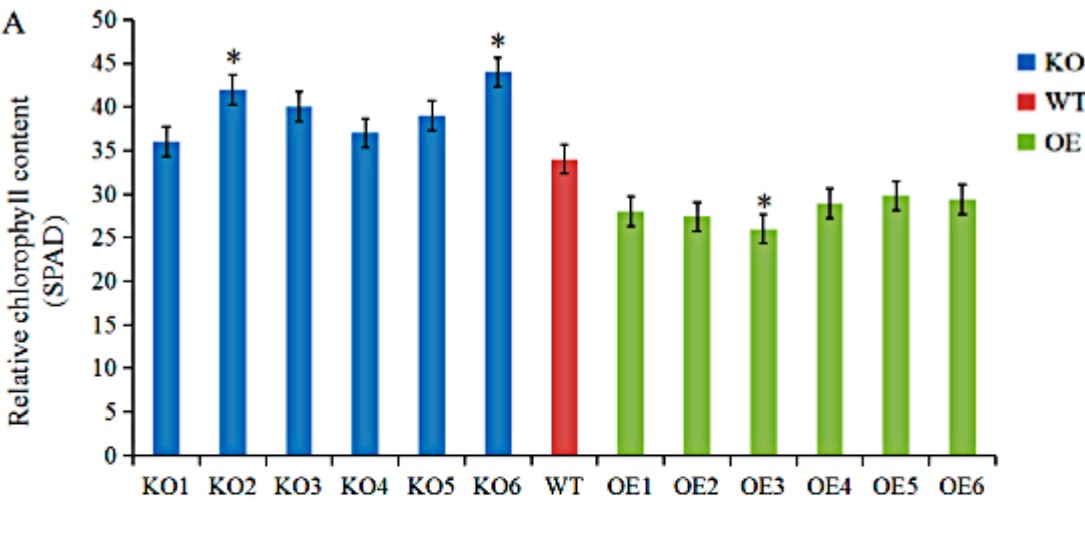

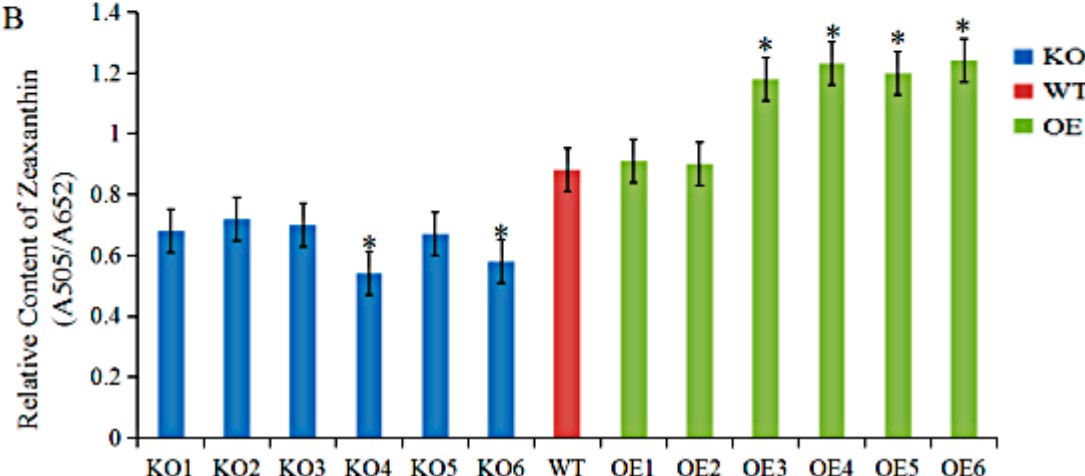

**Figure 5.** Non-photochemical quenching (NPQ)-related physiological indicators of WT and the $T_2$ generation of *GmRIQ2-like*-OE and -KO transgenic one-month seedlings. (**A**) SPAD (soil-plant analyses development) values of WT and *GmRIQ2-like*-OE transgenic soybean seedlings were detected using a SPAD-502 Chl meter to obtain the relative contents of Chl. (**B**) The relative zeaxanthin contents of WT, *GmRIQ2-like*-OE, and -KO transgenic soybean seedlings were detected using a spectrophotometer at 505 and 652 nm. Vertical bars represent the SE ($n = 3$; each treatment consisted of 30 seedlings). Asterisks indicate significant differences (* $p < 0.05$) between the transgenic lines and WT.

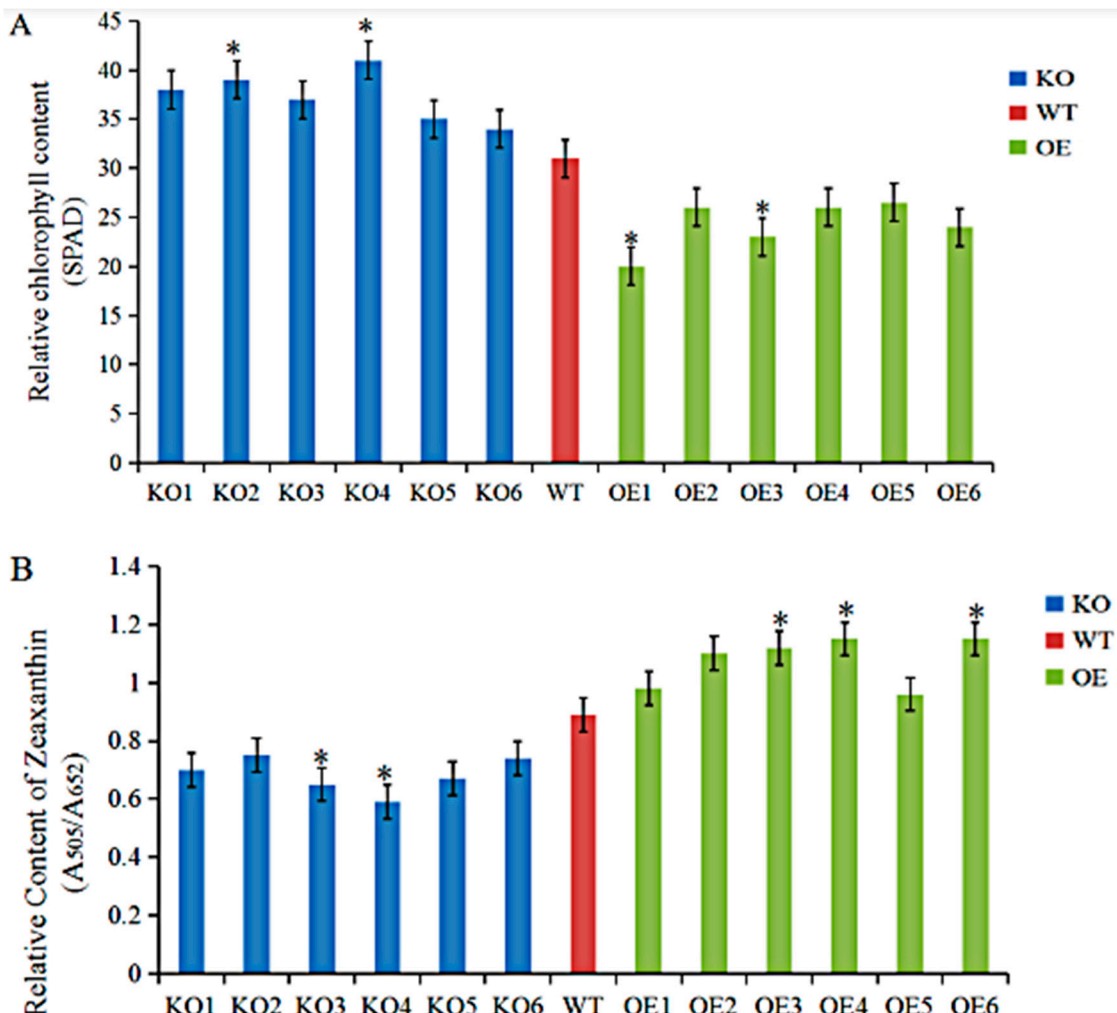

**Figure 6.** NPQ-related physiological indicators of WT and the $T_3$ generation of *GmRIQ2-like*-OE and -KO transgenic one-month seedlings. (**A**) SPAD values of WT and *GmRIQ2-like*-OE transgenic soybean seedlings were detected using a SPAD-502 Chl meter to obtain the relative contents of Chl. (**B**) The relative zeaxanthin contents of WT, *GmRIQ2-like*-OE, and -KO transgenic soybean seedlings were detected using a spectrophotometer at 505 and 652 nm. Vertical bars represent the SE ($n$ = 3; each treatment consisted of 30 seedlings). Asterisks indicate significant differences (* $p$ < 0.05) between the transgenic lines and WT.

### 3.5. Chl fluorescence Kinetic Parameters of the $T_2$ and $T_3$ Generations of GmRIQ2-like Transgenic Seedlings

Changes in the Chl fluorescence parameters, Fo, Fm, Fv/Fm, and NPQ in the $T_2$ and $T_3$ generations of *GmRIQ2-like*-OE, -KO, and WT seedling leaves under different light intensities were measured (Figures 7 and 8). Fo represents the fluorescence yield of the PS II RC in the completely open state, and changes in its value infer the state of the RC. In the $T_2$ generation of *GmRIQ2-like*-OE, -KO, and WT seedlings, Fo was ranked as follows: *GmRIQ2-like*-OE < WT < *GmRIQ2-like*-KO (Figure 7A). These results indicated that the photosynthetic electron transport of the PS II RC in soybean leaves was blocked due to strong light stress, which resulted in damage to the PS II RC. Fv/Fm of the $T_2$ generation of *GmRIQ2-like*-OE, -KO, and WT seedlings was ranked as follows: *GmRIQ2-like*-OE > WT > *GmRIQ2-like*-KO (Figure 7B). Compared to *GmRIQ2-like*-KO and WT seedlings, *GmRIQ2-like*-OE seedlings exhibited a stronger heat dissipation ability under strong light stress. The gradual enhancement of light intensity led to a gradual increase of NPQ in the $T_2$ generation of *GmRIQ2-like*-OE, -KO, and WT seedlings, which were ranked as follows: *GmRIQ2-like*-OE > WT >

*GmRIQ2-like*-KO (Figure 7C). These results further indicated that the electron transfer of PS II was inhibited, and this ability decreased due to strong light stress, which thereby led to a decrease in photosynthetic capacity. The heat dissipation of NPQ dissipated excess light energy and protected photosynthetic organs that were not destroyed by photoinhibition, which thereby alleviated the effects of strong light stress on photosynthesis. The NPQ of *GmRIQ2-like*-OE seedlings increased significantly, which improved their heat dissipation ability and strong light stress resistance, thereby alleviating photoinhibition and playing a role in photoprotection. When the $T_3$ generation of *GmRIQ2-like*-OE, -KO, and WT seedlings was resistant to strong light, changes in the trends of various parameters were roughly the same as the $T_2$ generation seedlings (Figure 8).

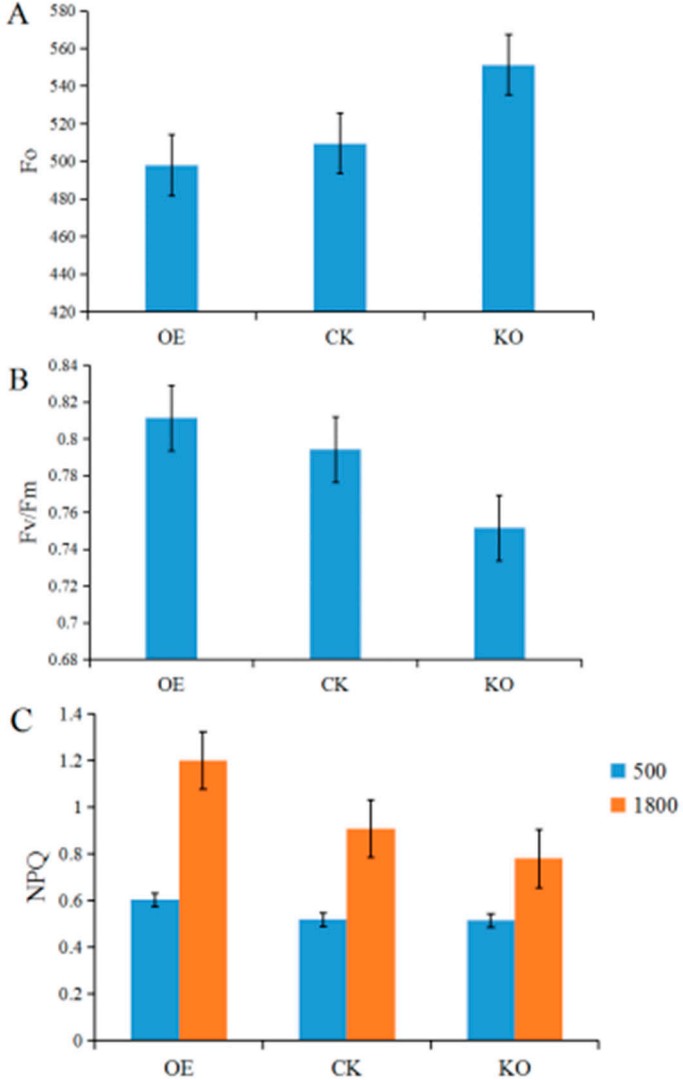

**Figure 7.** Chl fluorescence kinetic parameters of WT and the $T_2$ generation of *GmRIQ2-like*-OE and -KO transgenic soybean seedlings. (**A–C**) represent the effects of different light intensities on the Fo, Fv/Fm, and NPQ of three types of soybean leaves. Vertical bars represent the SE ($n = 3$; each treatment consisted of 30 seedlings).

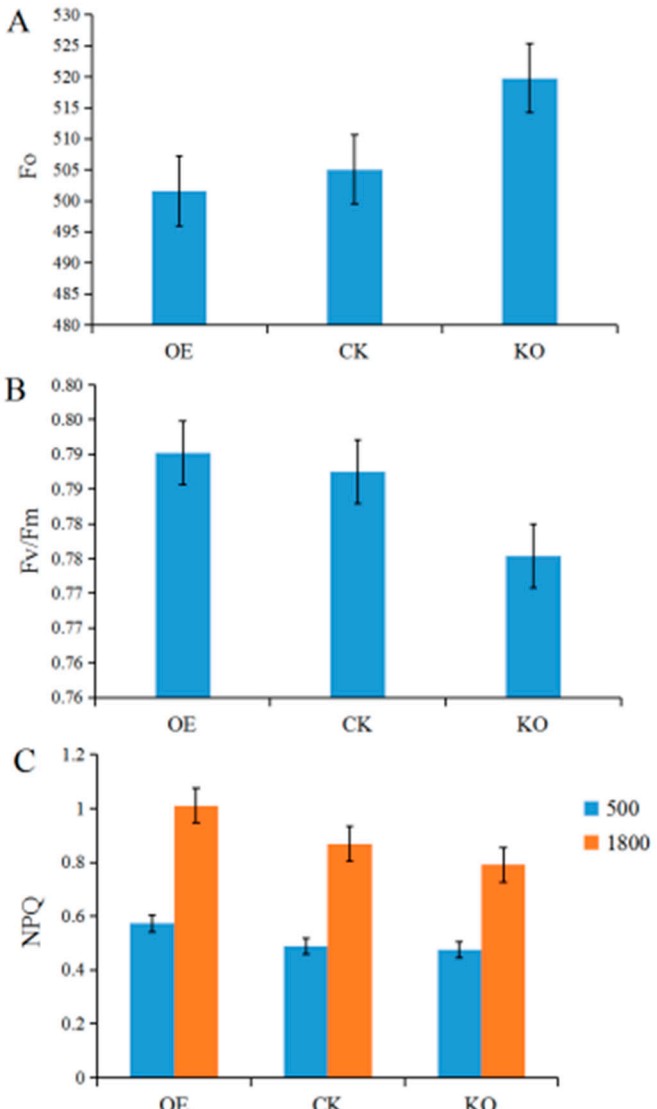

**Figure 8.** Chl fluorescence kinetic parameters of WT and the $T_3$ generation of *GmRIQ2-like*-OE and -KO transgenic soybean seedlings. (**A–C**) represent the effects of different light intensities on the Fo, Fv/Fm, and NPQ of three types of soybean leaves. Vertical bars represent the SE ($n$ = 3; each treatment consisted of 30 seedlings).

The Chl fluorescence imager described the Chl fluorescence parameters of the $T_2$ and $T_3$ generations of *GmRIQ2-like*-OE, -KO, and WT seedling leaves. The brightness of the NPQ image decreased significantly under strong light stress and the brightness change in the *GmRIQ2-like*-OE image was the most significant for both generations (Figures 9 and 10).

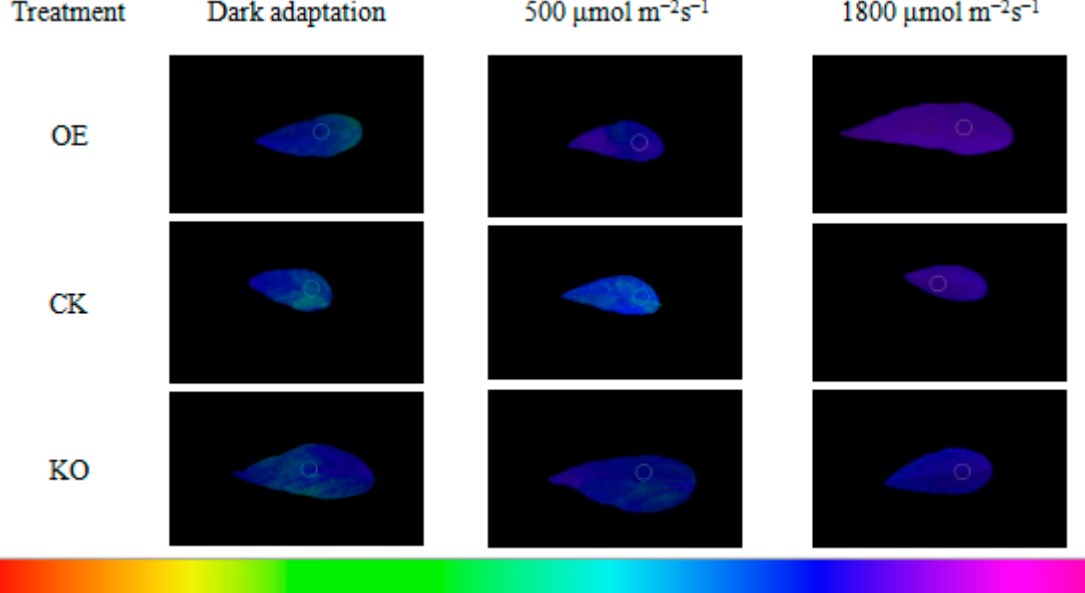

**Figure 9.** Effects of strong light stress on Chl fluorescence images of WT and the $T_2$ generation of *GmRIQ2-like*-OE and -KO transgenic soybean seedlings.

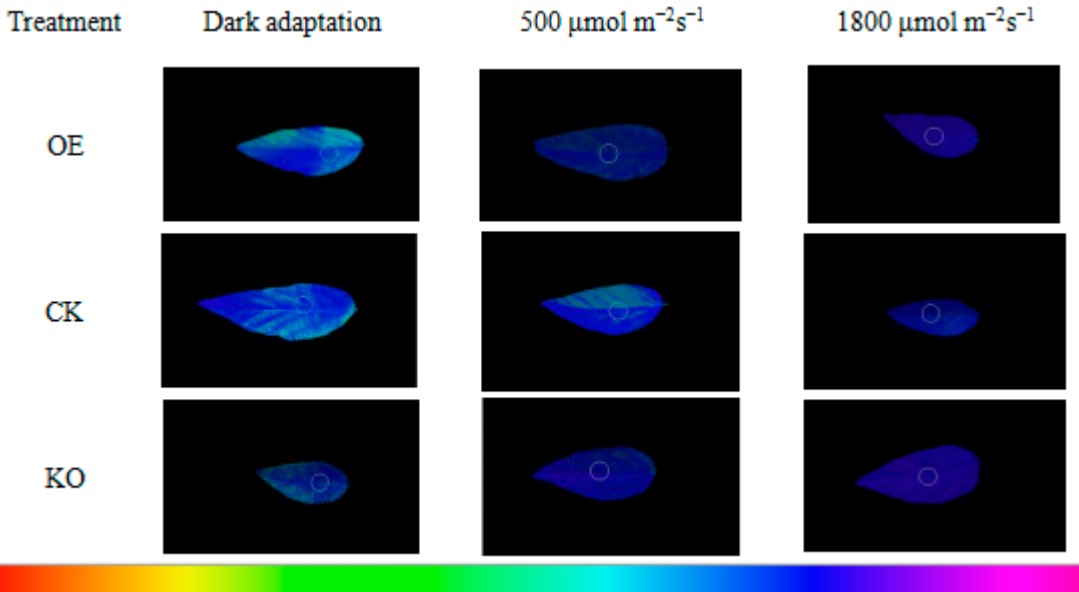

**Figure 10.** Effects of strong light stress on Chl fluorescence images of WT and the $T_3$ generation of *GmRIQ2-like*-OE and -KO transgenic soybean seedlings.

## 4. Discussion

### 4.1. Cloning and Functional Analysis of GmRIQ2-like

Grana are stacked thylakoid membrane structures found in land plants that contain PS II and LHC IIs [65]. *AtRIQ2* encodes uncharacterized grana thylakoid proteins, which are small proteins localized in the grana margin and stromal lamella [42]. Compared to the *AtRIQ2* protein sequence, *GmRIQ2-like* was cloned successfully from the soybean cDNA library. Grana stacking is well-developed in land plants, and *RIQ* genes are highly conserved [66]. Therefore, the localization of *GmRIQ2-like* in this study was consistent with *AtRIQ2*. Based on the homology analysis, *GmRIQ2-like* was highly homologous with *P. vulgaris* L. In plants with artificially modulated LHC II levels, the degree of stacked grana

depended on the LHC II levels [43,46]. RIQ proteins directly regulated the extent of grana stacking and LHC II organization around PS II, which was consistent with the phylogenic distribution of RIQ genes in phototrophs that develop the grana structure [42]. As a result, the RIQ proteins optimized LHC II organization, adjusted the photoprotective functions, and increased the sensitivity of LHC IIs to ΔpH by optimizing the grana conditions [67]. Moreover, the activation of NPQ was dependent on the formation of proton gradients across thylakoid membranes (DpH) [68]. Thus, *GmRIQ2-like* clearly increased the NPQ of soybeans.

### 4.2. Yield Variation between GmRIQ2-like-OE, -KO, and WT Soybean Seedlings

Light is one of the key elements of photosynthesis that affects plant growth and development [7]. Under intense light stress, excess light significantly reduces the electron transport rate, the accumulation of reactive oxygen species seriously damages RCs, and the photosynthetic activity and photochemistry efficiency are reduced, resulting in photoinhibition [69,70]. Therefore, it is necessary for plants to activate NPQ to dissipate excess light energy as heat. This phenomenon is widespread in plants and characterized by the reduction of photochemical efficiency, Fv/Fm, and photosynthetic carbon assimilation of PS II [16]. For example, *Citrus unshiu* leaves are affected by heat and strong light under sunny summer conditions. These stressors cause the stomatal closure of leaves, limit $CO_2$ assimilation, result in excess energy, reduce electron transport chain continuity, and increase peroxide accumulation. [71]. In this study, the promoter element analysis indicated that *GmRIQ2-like* participated in photosynthesis and defense and stress responses of several abiotic stressors. Inferred from the results, as a gene that may advance NPQ, *GmRIQ2-like* protected soybean seedlings from strong light stress by dissipating excess activation energy as heat. Consistently, under the tested conditions, the agronomic trait results revealed that the photosynthetic products were yields of *GmRIQ2-like*-OE, -KO, and WT soybean seedlings, which were ranked as follows: *GmRIQ2-like*-KO > WT > *GmRIQ2-like*-OE. The results are not significantly different at the level of * $p < 0.05$, but can reflect a certain trend.

### 4.3. Energy Dissipating Ability Comparison between GmRIQ2-like-OE and WT Seedlings

Previous studies demonstrated that greater reductions in the quantum efficiency of PS II photochemistry were attributed to increased energy dissipation [72]. Under strong light stress, substantial damages to photosynthetic pigments and decreased Chl contents were commonly observed phenomena [73,74]. Chl fluorescence is a useful, non-invasive tool used for the study of different aspects of photosynthesis, as well as for the detection of various environmental stressors in a wide range of plant species [75]. Moreover, SPAD values are positively correlated with Chl content [76]. In a previous study, *Arabidopsis* protected itself from short-term stress by increasing its NPQ level, which thereby dissipated light energy and decreased the efficiency of photochemical reactions in photosynthesis [77]. In this study, under the tested conditions, the SPAD values of *GmRIQ2-like*-OE seedlings were lower than WT. These results were consistent with the functions of *GmRIQ2-like*, which boosted energy dissipation and depressed Chl synthesis.

### 4.4. Xanthophyll Cycle Pool Analysis of GmRIQ2-like-OE, -KO, and WT Soybean Seedlings

The results indicated that zeaxanthin is important in the dissipation of excessive energy when plants are subjected to environmental stress factors in the presence of light [10]. Increased zeaxanthin contents or xanthophyll cycle pool sizes, which allow for the potential formation of zeaxanthin, were presumably related to a greater capacity to dissipate excess excitation energy as heat [10]. The xanthophyll cycle is involved in the conversion of violaxanthin to zeaxanthin via antheraxanthin by deoxycyclase [78]. In this study, under the tested conditions, zeaxanthin contents of *GmRIQ2-like*-OE soybean seedlings were greater than WT, and WT contents were greater than *GmRIQ2-like*-KO soybean seedlings. This finding was in accordance with the above results. A previous study found that the *npq2* mutant blocked in the conversion of zeaxanthin to violaxanthin accumulates zeaxanthin and has a higher NPQ than WT cells [79]. In the presence of several abiotic stressors, sustained yield

reductions were associated with both sustained increases in thermal energy dissipation (associated with zeaxanthin) and zeaxanthin contents [10]. Under water stress when watering was terminated, *Nerium oleander* leaves, which do not perform osmotic adjustment, exhibited sustained decreases in Chl fluorescence yields, indicating increased thermal energy dissipation and sustained increases of zeaxanthin levels [80]. Under heat stress, the rate of de-epoxidation of violaxanthin to zeaxanthin increased as leaf temperature increased, which is expected of an enzymatic reaction [81]. A similar temperature response was observed for the development of thermal energy dissipation based on the changes in Chl fluorescence yield of intact leaves [82]. Under cold stress in exposed field habitats, several evergreen species contained large zeaxanthin contents prior to sunrise during cold periods in the winter [83]. The retention of zeaxanthin was also accompanied by sustained reductions in photochemical efficiency [84].

In this study, the tissue culturing technique of soybean cotyledonary nodes was employed to obtain transgenic soybean seedlings. Additionally, the advanced CRISPR/cas9 gene KO technique was applied, which led to a high *GmRIQ2-like* silencing efficiency. Through various analyses of transgenic seedlings, which were used for determining photosynthesis and NPQ pathway products, results revealed that the main function of *GmRIQ2-like* was the enhancement of soybean NPQ to dissipate excess light energy and reduce or prevent plant photoinhibition in order to protect the plant. These findings will serve as a theoretical basis for the molecular breeding of soybean resistance, especially to strong light stress, and provide a foundation for cultivating a wide range of new soybean germplasms.

## 5. Conclusions

A soybean gene, *GmRIQ2-like*, was found to be highly homologous with *AtRIQ2* in this study. *GmRIQ2-like* was localized in the grana and promoted NPQ. An assay of transgenic *GmRIQ2-like*-OE and -KO soybean seedlings revealed that *GmRIQ2-like* improved the photoinhibition sensitivity of plants to strong light by enhancing the dissipation of excess energy. Additionally, the agronomic traits and relative Chl and zeaxanthin contents also indicated that *GmRIQ2-like* resisted the degree of photoinhibition in soybeans. These results provide additional evidence that RIQ family genes function during photoinhibition and substantially increase our understanding of the molecular mechanisms of strong light tolerance. Moreover, these findings will serve as a reference for future studies on the responses of strong light-related pathways in soybeans and other crops.

**Author Contributions:** Conceptualization, B.Z.; methodology, H.Y., L.M. and J.Z.; software, J.Z.; validation, J.D.; formal analysis, D.L. and B.Z.; investigation, H.Y., L.M. and J.Z.; resources, B.Z.; data curation, J.D., D.L. and B.Z.; writing—original draft preparation, J.D.; writing—review and editing, J.D. and B.Z.; visualization, J.D., D.L. and B.Z.; supervision, D.L. and B.Z.; project administration, D.L.; funding acquisition, B.Z. All authors have read and agreed to the published version of the manuscript.

**Funding:** This research was funded by Breeding of New Soybean Varieties with Antireversing Genes No. 2016ZX08004002; Postdoctoral Scientific Research Developmental Fund of Heilongjiang Province, grant number No. LBH-Q18024 and "Cloning and preliminary function analysis of soybean GmRIQ2 gene" project, grant number No. SB17A02.

**Conflicts of Interest:** The authors declare no conflict of interest. The funders had no role in the design of the study; in the collection, analyses, or interpretation of data; in the writing of the manuscript, or in the decision to publish the results.

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
