# Peer review of "Overexpression of GmRIQ2-like (Glyma.04G174400) Enhances the Tolerance of Strong Light Stress and Reduces Photoinhibition in Soybean (Glycine max (L.) Merr.)"

_agriculture, doi:10.3390/agriculture10050157_

Round 1

Reviewer 1 Report

Dear authors,

I read with great pleasure your manuscript. It is always good to learn about a new gene that has been described and functionally characterized.

The work is nicely done and the essential data is presented. I have made only a few comments and suggestions bellow:  

Line 17: “was highly homogeneous” would be better replaced by “had highly similar sequence to

or with high similarity to”

Line 98: When presenting the cultivars please use the "full names" of the cultivars at least once, when it is the case. Specify the reason for the choice of such cultivars.  

Line 146: under is repeated twice

Line 161: It is not clear from this point onwards in which condition the plants used for phenotyping were grown. Was it in pots or field? What is the photoperiod? what is the light intensity or wave length characteristics at the moment of the measurements?

Line 177: seedlings should be replaced by plants

Lines 247 and 248: Please check the seedlings numbering

Line 255: About the ranking of the agronomical traits. Considering the SE I am not confident there is a clear or real difference there.

Line 265: “under normal light conditions...” I see that only standard conditions were tested in the present work so better to mention only, "under the tested conditions". Every time I read control or normal it rises expectations for results in non-ideal or stress conditions.

Line 271: “ultimately affected soybean yield” This is a very strong statement considering the SE. I don't see statistics so I wouldn't make such strong affirmation. Besides, there is no information about total seed production in weight, which is a great indicator of yield and could be a more confident way to inform about yield as the seed size could have been affected.

Line 295: please check figure number

Line 357: there is a strong conclusion about the effect on agronomical traits. As mentioned before, I don't think there is clear difference in yield based on the results presented. I would suggest growing the plants also in higher light conditions to draw a conclusion about the effect of GmRIQ2-like expression on agronomic traits.

I would like to remark that it would have added a lot for the agronomical/breeding community to include experiments under stressful light conditions.

Your sincerely

Author Response

Response to Reviewer 1 Comments

We would like to thank the reviewer for their careful reading and thoughtful comments on our previous draft. We have taken your comments into careful consideration while preparing this revision. We have carefully answered the questions one by one as required by yours and carefully revised the article.We summarize our responses to your comments below.

According to the requirements of the instructions, we have changed the order of manuscripts, and we have marked the corresponding lines below.

Point 1: Line 17: “was highly homogeneous” would be better replaced by “had highly similar sequence to or with high similarity to”.

Response 1: Our expressions are not accurate enough, and the suggestions of the reviewer can describe our thoughts more accurately. The original sentence "was highly homogeneous" has been changed to "had highly similar sequence to".

Point 2: Line 98: When presenting the cultivars please use the "full names" of the cultivars at least once, when it is the case. Specify the reason for the choice of such cultivars.

Response 2: Now Line 320: The full names of "KF16" and "DN50" are "KenFeng 16" and "DongNong 50", respectively. This is partly caused by our negligence. Corrections have now been made. Thanks to the reviewer for his careful reminder. We use such cultivars for the following reasons: Soybean is a crop with low genetic transformation efficiency. After we tried to transform, we found that the transformation efficiency of "DongNong 50" and "KenFeng 16" was higher than that of other cultivars, which was beneficial to our subsequent research. And "KenFeng 16" is the main cultivar in Heilongjiang Province, China. So we choose such cultivars for research.

Point 3: Line 146: under is repeated twice.

Response 3: Sorry we didn't find a place to repeat twice. Not sure if it's a typographical error. If there is an opportunity, could you please elaborate?

Point 4: Line 161: It is not clear from this point onwards in which condition the plants used for phenotyping were grown. Was it in pots or field? What is the photoperiod? what is the light intensity or wave length characteristics at the moment of the measurements?

Response 4: Now Line 401: Both our transgenic and WT plants were grown in soil: vermiculite (1:1) pots under a 16/8 h light/dark photoperiod at 23°C. The light intensity was maintained at about 500 μmol m-2s-1 during the measurement.

Point 5: Line 177: seedlings should be replaced by plants.

Response 5: Now Line 401: After a reviewer's reminder, we think that the plants are more appropriate than the seedlings. The original "seedings" was changed to "plants".

Point 6: Lines 247 and 248: Please check the seedlings numbering.

Response 6: Lines 247 and 248: Now Line 131 and Line 132: the content of the description picture has been ignored and has now been changed.

Point 7: Line 255: About the ranking of the agronomical traits. Considering the SE I am not confident there is a clear or real difference there.

Response 7: Now Line 144 and Line 158: Regarding the ranking of agronomic traits, the questions raised by the reviewer are necessary. We supplemented the relevant data, including Weight of seed per plant and Hundred-grain weight (Figures 4A and 4B). This will make the manuscript look more complete.

Point 8: Line 265: “under normal light conditions...” I see that only standard conditions were tested in the present work so better to mention only, "under the tested conditions". Every time I read control or normal it rises expectations for results in non-ideal or stress conditions.

Response 8: After careful review of the manuscript, we consider the reviewer's description more rigorous. The experimental conditions in the manuscript have been corrected. Line 142、Line 247、Line 260 and Line 269: The original sentence "under the tested conditions" is changed to "under control conditions". Line 150: The original sentence "under normal light conditions" is changed to "under the tested light conditions".

Point 9: Line 271: “ultimately affected soybean yield” This is a very strong statement considering the SE. I don't see statistics so I wouldn't make such strong affirmation. Besides, there is no information about total seed production in weight, which is a great indicator of yield and could be a more confident way to inform about yield as the seed size could have been affected.

Response 9: Now Line 155: We supplemented the data on agronomic traits, including Weight of seed per plant and Hundred-grain weight. According to the results, the connection between expression of transgenic seedlings and soybean yield under the tested conditions can be inferred. Our representations in the manuscript were not clear enough and have now been corrected.

Point 10: Line 295: please check figure number.

Response 10: Now Line 181: The figure number has changed. The "Fig. 5A" has been changed to "Fig. 7A". Thanks to the reviewer for hisr careful reading.

Point 11: Line 357: There is a strong conclusion about the effect on agronomical traits. As mentioned before, I don't think there is clear difference in yield based on the results presented. I would suggest growing the plants also in higher light conditions to draw a conclusion about the effect of GmRIQ2-like expression on agronomic traits.

Response 11: Now Line 244: Regarding the expression of the effects of agronomic traits, we can concluded that the expression of GmRIQ2-like had a negative effect on soybean yield under the tested conditions(Line 142). As you said, the results did not reach a significant difference at the statistical level of p <0.05. It may be that our description is not accurate enough and has now been corrected. However, under the tested conditions, a certain trend can be reflected.In addition, our research is still in progress. Your suggestions will be discussed and pre-tested in subsequent studies. Thank you again for your valuable suggestions to our manuscript.

Reviewer 2 Report

Dear Editor, Plants:

Please receive my review of the manuscript entitled “ Overexpression of GmRIQ2-like (Glyma.04G174400) enhances the tolerance of strong light stress and reduces photoinhibition in (Glycine max L. Merr.)” by Deng et al. 2020.

The manuscript is well written. The authors were able to over-express GmRIQ2-like (Glyma.04G174400) to enhance the tolerance of strong light stress and reduce photoinhibition in soybean. They showed overexpression of GmRIQ2-like- using the quantitative real-time (qRT)-PCR and demonstrated that GmRIQ2-like dissipated excess light energy by enhancing the non-photochemical quenching and reduced plant photoinhibition. They concluded that the resulted transgenic soybeans were able to over-express GmRIQ2-like and used to enhance strong light stress tolerance and protect soybean plants from photoinhibition damage. This study will further expand our understanding of photoprotection regulation mechanisms. My comments to improve the manuscript are below. Please let me know if you have questions.

My comments to improve the manuscript are below:

-The authors need to re-order the manuscript main sections: Abstract, Introduction, Results, Discussion, Materials and Methods, and so on. The authors may visit the instructions to authors for manuscript preparation.

-The authors may change “Fig.” in text to “Figure” according to the journal instructions.

-The authors may need to improve the quality and resolution of all Figures (graphs).

- Revision throughout the manuscript is also recommended for possible improvement for better clarity.

Author Response

Response to Reviewer 2 Comments

We would like to thank the reviewer for their careful reading and thoughtful comments on our previous draft. We have taken your comments into careful consideration while preparing this revision. We have carefully answered the questions one by one as required by yours and carefully revised the article.We summarize our responses to your comments below.

Point 1: The authors need to re-order the manuscript main sections: Abstract, Introduction, Results, Discussion, Materials and Methods, and so on. The authors may visit the instructions to authors for manuscript preparation.

Response 1: We have rearranged the order of manuscripts in accordance with the instructions and the reviewers' suggestions.

Point 2: The authors may change “Fig.” in text to “Figure” according to the journal instructions.

Response 2: It has been changed from "Fig." to "Figure" in the manuscript according to the journal instructions.

Point 3: The authors may need to improve the quality and resolution of all Figures (graphs).

Response 3: We re-uploaded the figures. Improve the quality and resolution of all figures as much as possible.

Point 4: Revision throughout the manuscript is also recommended for possible improvement for better clarity.

Response 4: We have comprehensively revised and improved the manuscript. Thank you for your valuable suggestions.
